# Low Blood-As Levels and Selected Genotypes Appears to Be Promising Biomarkers for Occurrence of Colorectal Cancer in Women

**DOI:** 10.3390/biomedicines9091105

**Published:** 2021-08-28

**Authors:** Piotr Baszuk, Paulina Stadnik, Wojciech Marciniak, Róża Derkacz, Anna Jakubowska, Cezary Cybulski, Tomasz Huzarski, Jacek Gronwald, Tadeusz Dębniak, Katarzyna Białkowska, Sandra Pietrzak, Józef Kładny, Rodney J. Scott, Jan Lubiński, Marcin R. Lener

**Affiliations:** 1International Hereditary Cancer Center, Department of Genetics and Pathology, Pomeranian Medical University in Szczecin, ul. Unii Lubelskiej 1, 71-252 Szczecin, Poland; piotr.baszuk@pum.edu.pl (P.B.); paulina.z.stadnik@gmail.com (P.S.); wojciech.marciniak@read-gene.com (W.M.); roza.derkacz@gmail.com (R.D.); aniaj@pum.edu.pl (A.J.); cezarycy@pum.edu.pl (C.C.); huzarski@pum.edu.pl (T.H.); jgron@pum.edu.pl (J.G.); debniak@pum.edu.pl (T.D.); katarzyna.kaczm@gmail.com (K.B.); sandra.pietrzak@pum.edu.pl (S.P.); marcinlener@poczta.onet.pl (M.R.L.); 2Read-Gene, Grzepnica, ul. Alabastrowa 8, 72-003 Grzepnica, Dobra (Szczecińska), Poland; 3Department of Clinical Genetics and Pathology, University of Zielona Góra, ul. Zyty 28, 65-046 Zielona Góra, Poland; 4Department of General Surgery and Surgical Oncology, First Clinical Hospital of Pomeranian Medical University in Szczecin, ul. Unii Lubelskiej 1, 71-252 Szczecin, Poland; jkladny@onet.pl; 5Priority Research Centre for Cancer Research, Innovation and Translation, Hunter Medical Research Institute, New Lambton Heights, NSW 2308, Australia; rodney.scott@newcastle.edu.au; 6School of Biomedical Sciences and Pharmacy, Faculty of Health and Medicine, University of Newcastle, Callaghan, NSW 2308, Australia; 7Division of Molecular Medicine, Pathology North, John Hunter Hospital, New Lambton, NSW 2305, Australia

**Keywords:** colorectal cancer, arsenic level, biomarkers

## Abstract

In following study we examined whether blood arsenic (As) levels combined with specific polymorphisms in *MT1B*, *GSTP1*, *ABCB1*, *NQO1*, *CRTC3*, *GPX1*, *SOD2*, *CAT*, *XRCC1*, *ERCC2* can be used as a marker for the detection of colorectal cancer (CRC) among Polish women. A retrospective case-control study of CRC included 83 CRC cases and 78 healthy controls. From each study participant pre-treatment peripheral blood was collected for As level measurement by inductively coupled–plasma mass spectrometry (ICP-MS). We estimated the odds ratio (OR) of the association between blood-As levels and CRC using multivariable unconditional logistic regression models. A low blood-As level (0.27–0.67 µg/L) was associated with an increased frequency of CRC (OR: 3.69; *p* = 0.005). This correlation was significantly greater when participants carried particular gene variants: *CAT*, rs1001179-nonCC (OR: 19.4; *p* = 0.001); *ABCB1* rs2032582–CC (OR: 14.8; *p* = 0.024); *GPX1* rs1050450-CC (OR: 11.6; *p* = 0.002) and *CRTC3* rs12915189-nonGG (OR: 10.3; *p* = 0.003). Our study provides strong evidence that low blood-As levels are significantly associated with increased CRC occurrence and that particular gene variants significantly enhanced this correlation however, due to the novelty of these findings, we suggest further validation before a definitive statement that the combined effect of low blood-As levels with specific gene polymorphisms is a suitable CRC biomarker.

## 1. Introduction

There are 2 major methods to study association between biomarkers and cancer. The first one studies the risk of cancer development and is based on a prospective cohort when patients are initially unaffected. The other method, a case-control study explores cancer occurrence. Although the case-control study design has limitations as it does not exclude influence of tumour itself on biochemical parameters, this type of study is very useful to establish the most efficient criteria of cancer surveillance. Malignant tumours are the leading cause of death worldwide [1,2]. According to GLOBOCAN 2020 data, colorectal cancer (CRC) was the third most frequently diagnosed cancer and was classified as one of the most common types of cancers in both men and women [3]. It is believed that the interaction of various risk factors cause the accumulation of many genetic/epigenetic/metabolic changes within the cell has the greatest impact on cancer initiation and development [4,5,6,7,8,9]. It is known that low concentrations of some trace elements is beneficial for human health, however it has also been shown that genetic polymorphisms that alter the function or expression of important genes that play key roles in the metabolism of trace elements may contribute to the increased risk of cancer development [10,11]. Moreover, several trace elements are considered as chemical carcinogens that may contribute to the carcinogenic process [10,11]. The IARC report determined arsenic (As) and its compounds as one of the most carcinogenic trace elements that mankind is exposed to [10,12,13]. Although As is ubiquitous in the environment, and human exposure can occur from many natural and anthropogenic sources, diet is classified as one of the major source of As exposure [14]. Biomolecular mechanisms of As clearance are still poorly understood, however there are some theories that its biotransformation may exert carcinogenesis in both a genotoxic and non-genotoxic manner [15,16]. Recently we have been able to show that As is a major cancer risk factor for women in Poland–high levels of blood As (the highest quartile) is associated with more than 10-times higher risk of breast cancers as compared to quartile with the lowest As levels [17]. The above correlations have not been observed in our studies of men (data not shown). So far many retrospective revisions have indicated an association between the concentration of As in human samples and the occurrence of various cancer types [18,19,20,21,22,23,24,25,26], however none of these reports studied the relationship between As exposure and CRC occurrence. It is well established that increased risks of cancer are associated with polymorphisms of different genes. For the studies herein, we selected polymorphisms in genes reported to be directly involved in malignant transformation and/or xenobiotic metabolism and/or oxidative stress. As some studies have already explored the combined effects of metal levels with DNA variants of genes involved in carcinogenesis [27,28], we decided to explore whether functional polymorphisms in a panel of genes selected by us correlates with the frequency of CRC linked with blood-As levels in Polish women. Based on functional characteristics, high mutant-allele frequency (MAF > 25%) and potential relationship with cancer development and occurrence, 10 gene polymorphisms were studied. We selected the following genes: ATP Binding Cassette Subfamily B Member 1 (*ABCB1*) [29,30] and glutathione s-transferase P1 (*GSTP1*) [31] participating in elimination of some potentially toxic xenobiotics excision repair cross-complementing rodent repair deficiency group 2 (*ERCC2*) [32] and X-ray repair cross-complementing protein 1 (*XRCC1*) [33,34] that are involved in DNA repair, catalase (*CAT*), glutathione peroxidase-1 (*GPX1*), superoxide dismutase 2 (*SOD2*) [35] and NAD(P)H Quinone Dehydrogenase 1 (*NQO1*) [36] that are anti-oxidative stress enzymes, CREB Regulated Transcription Coactivator 3 (*CRTC3*) that plays a role in a regulation of CREB-dependent gene transcription in a phosphorylation-independent manner [37], metallothionein 1B (*MT1B*) which is involved in metal homeostasis and protection against heavy metal toxicity, DNA damage, and oxidative stress [38]. The aim of this study was to answer the following questions: Can blood-As levels be used as a biomarker for CRC occurrence in Polish women? Can specific gene polymorphisms be used as a biomarker for CRC occurrence in Polish women? Does the combined effect of blood-As levels and specific gene polymorphisms influence CRC frequency in Polish women?

## 2. Materials and Methods

### 2.1. Study Group

A population based retrospective case-control study of CRC was conducted on the cohort of 161 Polish women. The characteristics of cases and control participants are presented in Table 1. All study subjects provided written informed consent to participate in the study and were randomly included in the study between 2012 and 2017. For each cancer patient, one unaffected individual registered at the International Hereditary Cancer Centre, Pomeranian Medical University of Szczecin, was matched as a healthy control. Participants were matched for year of birth (±3 years), sex, smoking status (pack-years ±20%) and the total number of malignancies among first degree relatives. In all cases cancer was confirmed by histopathological examination. The study was conducted in accordance with the Helsinki Declaration and with the consent of the Ethics Committee of Pomeranian Medical University in Szczecin under the number KB-0012/73/10.

### 2.2. Measurement of Blood-As Level

10 mL of peripheral blood was collected into a vacutainer tube containing sodium ethylenediaminetetraacetic acid (EDTA) from all study participants and then stored at −80 °C until the day of analysis. Study subjects were fasting for at least six hours before sample collection. Blood samples were taken from patients’ pre-treatment at the time of diagnosis. Total arsenic concentration in their blood was measured by the inductively coupled plasma mass spectrometer method reported in our previous study [17].

### 2.3. Molecular Analysis

Ten selected variants in ten genes were genotyped: rs13181 in *ERCC2*, rs1799782 in *XRCC1*, rs7191779 in *MT1B*, rs1695 in *GSTP1*, rs2032582 in *ABCB1*, rs1800566 in *NQO1*, rs12915189 in *CRTC3*, rs1050450 in *GPX1*, rs4880 in *SOD2*, and rs1001179 in *CAT*. From each individual included in the study, a 10 mL peripheral blood sample was collected in a vacutainer tube containing 1 mL of 10% sodium EDTA. The genomic DNA was isolated using the detergent method [39]. SNP Analyses were performed using a pre-designed Genotyping Assay × 40 (Applied Biosystems, Waltham, Massachusetts, USA). Each reaction mixture consisted of 2.5 μL LightCycler 480 Probe Master Mix (Roche Diagnostics, Basel, Switzerland), the assay 0.125 μL (Genotyping Assay × 40 TaqMan) (Applied Biosystems, Waltham, Massachusetts, USA) and 1.375 μL deionized water (Roche Diagnostics, Basel, Switzerland). Samples were analyzed on 384-well plates. Each plate was included positive, negative and water-blind control. The genotyping data were collected and analyzed using the LightCycler 480 Instrument and the program of the LightCycler 480 Basic Software Version 1.5 (Roche Diagnostics, Basel, Switzerland).

### 2.4. Statistical Analysis

For the estimation of association of As levels or As level and genotype with female CRC occurrence, study participants were assigned to one of four categories (quartiles-Q) based on the As levels distribution among controls. The association of As level with CRC occurrence was estimated with odds ratio (OR) at 95% confidence intervals using multivariable unconditional logistic regression model. The fourth quarter was utilized as the reference category for the odds ratio calculation. All calculations were performed using: “R: A language and environment for statistical computing. R Foundation for Statistical Computing, Vienna, Austria” (R version 4.0.4 (10 October 2020)).

## 3. Results

The lowest blood-As level (0.27–0.67 µg/L–Q1) was associated with a 3.69-fold (95% CI 1.50–9.50); *p* = 0.005) increase in the risk of CRC when compared to blood As levels of 4th quartile (1.47–7.11 µg/L) (Table 2). Apart from that, significant differences were observed in As levels between controls and Stage I-II patients (median 0.9 and 0.66 respectively)–*p*-value = 0.01.

By comparing quartiles, further examination of the candidate gene variants showed a significantly greater probability of CRC occurrence in women with the lowest (0.27–0.67 µg/L) blood-As levels carrying functional polymorphisms such as: *CAT*, rs1001179-nonCC (OR: 19.4 (95% CI 3.69–159.0); *p* = 0.001) (Table 3), *ABCB1* rs2032582–CC (OR: 14.8 (95% CI 1.95–318); *p* = 0.024) (Table 4), *GPX1* rs1050450-CC (OR: 11.6 (95% CI 2.74–60.9); *p* = 0.002) (Table 5), *CRTC3* rs12915189-nonGG (OR: 10.3 (95% CI 2.42–52.8); *p* = 0.003) (Table 6). In the remaining 6 genes results were statistically non-significant or groups of cases/controls were very small.

None of the presented genotypes increased predisposition to CRC when blood-As level was not considered (Appendix A, Table A1, Table A2, Table A3 and Table A4).

## 4. Discussion

CRC is one of the major causes of morbidity and mortality in many well-developed countries and its occurrence is increasing in developing areas of the world.

This current retrospective report showed a novel finding that low blood-As levels are strongly associated with higher probability of CRC occurrence in women.

Since our published data achieved in our prospective cohort of women strongly suggested blood-As levels in women should be low, because it is correlated with an extremely low risk of all cancer development including CRC, we hypothesized that low blood-As levels in CRC patients (current retrospective study) may be due to significantly higher As metabolism occurring in cancerous cells during the growth of the CRC. Observed correlation may be also site-specific since in analogous studies performer in Our Centre, breast and lung cancer female patients did not show association between As level and tumour occurrence (data not shown). Mechanisms behind our observations need further investigations including somatic and genetic changes of various pathways involved in cancer development.

Recently, we recognized unexpectedly that one of mechanism of As induced carcinogenesis may be via estrogens. Experimental studies on animals have shown dependence of As effects on sex [40]. In prospective cohorts we also observed that As associated cancer risk was correlated with hormonal status: in women, higher blood As level was associated with higher cancer risk but in men, especially at age < 60 yrs, higher blood As level was associated with lower cancer risk (data not shown). In females, As may influence on development of cancers by disrupting the function of estrogen receptors and suppressing the signaling pathway of estrogen [41,42]. As is also a potential metallo-estrogen [43,44].

Surprisingly, to the best of our knowledge, there are no published retrospective case-control studies on the relationship between As levels in any biological sample and CRC occurrence.

It is well known that apart from many environmental factors, genetic predisposition plays a significant role in the CRC development and occurrence [45]. From the panel of selected genes in this study, we presented that gene variants such as: *CAT*, rs1001179-nonCC, *ABCB1* rs2032582–CC, *GPX1* rs1050450-CC, *CRTC3* rs12915189-nonGG significantly enhanced already greater CRC occurrence among women with ≤0.67 µg/L blood-As concentration, although none of these genotypes presented a significant effect on CRC occurrence on their own when blood-As levels were not considered. It is in agreement with results of former studies presenting no impact of polymorphisms of *ABCB1* rs2032582, *GPX1* rs1050450, *CAT* rs1001179 on CRC occurrence among respectively Asian [29], Norwegian [46], Iranian [47] German [48] and US populations [49].

As far as we are aware, this is the first report exploring the impact of *CRTC3* rs12915189 polymorphism on its own, but also in combination with blood-As levels on CRC occurrence. According to our findings, *CRTC3* rs12915189-nonGG itself did not predispose to a higher frequency of CRC, however with blood-As levels below or equal to 0.67 µg/L it significantly increased the occurrence of CRC among women from Poland.

Based on our data and assumptions, we consider that to exclude the occurrence of CRC, a colonoscopy should be offered to women with a low blood-As level as a result of either an As-free diet or found in their initial regular blood tests and in association with gene variants described above.

We are convinced that after further validation (also on larger numbers of cases and controls) the combination effects of low blood-As levels with *CAT* rs1001179-nonCC, *ABCB1* rs2032582–CC, *GPX1* rs1050450-CC, *CRTC3* rs12915189-nonGG maybe a suitable CRC biomarker in Polish women.

## 5. Conclusions

Low blood-As level (0.27–0.67 µg/L) significantly increases probability of the CRC occurrence in women.

*CAT* rs1001179-nonCC, *ABCB1* rs2032582–CC, *GPX1* rs1050450-CC, *CRTC3* rs12915189-nonGG polymorphisms enhance the already existing, increased probability of female CRC occurrence associated with low blood-As levels (0.27–0.67 µg/L).

## Figures and Tables

**Table 1 biomedicines-09-01105-t001:** Characteristics of female CRC cases and controls.

Total, *n* = 161	Participants	Age (Mean/Range)	Smoking Status		Stage of CRC				
			Yes	No	I	II	III	IV	Unknown
Cases	83	67.17 (35–90)	23	60	15	32	28	4	4
Controls	78	67.24 (36–88)	22	56	-	-	-	-	-

**Table 2 biomedicines-09-01105-t002:** CRC occurrence risk associated with As level in blood in women.

As Blood Level (µg/L)	Cases/Controls
Q1: 0.27–0.67	40/20
Q2: 0.68–0.88	11/19
Q3: 0.92–1.44	21/19
Q4: 1.47–7.11	11/20

Q1 vs. Q4 (reference), cases/controls: 40/20 vs. 11/20; OR: 3.69 (95% CI 1.50–9.50); *p* = 0.005.

**Table 3 biomedicines-09-01105-t003:** Numbers of CRC cases and healthy controls depending on *CAT* gene polymorphism and blood-As concentration in women.

*CAT* rs1001179				
As Blood Level (µg/L)	nonCC		CC	
	Cases	Controls	Cases	Controls
Q1: 0.27–0.67	20 (57%)	5 (17%)	20 (42%)	15 (31%)
Q2: 0.68–0.88	4 (11%)	7 (24%)	7 (15%)	12 (24%)
Q3: 0.92–1.44	9 (26%)	7 (24%)	12 (25%)	12 (24%)
Q4: 1.47–7.11	2 (5.7%)	10 (34%)	9 (19%)	10 (20%)

*CAT* rs1001179-nonCC: Q1 vs. Q4 (reference), OR: 19.4 (95% CI 3.69–159); *p* = 0.001.

**Table 4 biomedicines-09-01105-t004:** Numbers of CRC cases and healthy controls depending on *ABCB1* gene polymorphism and blood-As concentration in women.

*ABCB1* rs2032582				
As Blood Level (µg/L)	CC		nonCC	
	Cases	Controls	Cases	Controls
Q1: 0.27–0.67	14 (48%)	10 (31%)	26 (48%)	10 (22%)
Q2: 0.68–0.88	4 (14%)	7 (22%)	7 (13%)	12 (26%)
Q3: 0.92–1.44	10 (34%)	7 (22%)	11 (20%)	12(26%)
Q4: 1.47–7.11	1 (3.4%)	8 (25%)	10 (19%)	12(26%)

*ABCB1* rs2032582-CC: Q1 vs. Q4 (reference), OR: 14.8 (95% CI 1.95–318); *p* = 0.024.

**Table 5 biomedicines-09-01105-t005:** Numbers of CRC cases and healthy controls depending on *GPX1* gene polymorphism and blood-As concentration in women.

*GPX1* rs1050450				
As Blood Level (µg/L)	CC		nonCC	
	Cases	Controls	Cases	Controls
Q1: 0.27–0.67	18 (51%)	4 (12%)	22 (46%)	16 (36%)
Q2: 0.68–0.88	5 (14%)	8 (24%)	6 (12%)	11 (24%)
Q3: 0.92–1.44	6 (17%)	8 (24%)	15 (31%)	11 (24%)
Q4: 1.47–7.11	6 (17%)	13 (39%)	5 (10%)	7 (16%)

*GPX1* rs1050450-CC: Q1 vs. Q4 (reference), OR: 11.6 (95% CI 2.74–60.9); *p* = 0.002.

**Table 6 biomedicines-09-01105-t006:** Numbers of CRC cases and healthy controls depending on *CRTC3* gene polymorphism and blood-As concentration in women.

*CRTC3* rs12915189				
As Blood Level (µg/L)	GG		nonGG	
	Cases	Controls	Cases	Controls
Q1: 0.27–0.67	19 (40%)	15 (35%)	21 (58%)	5 (14%)
Q2: 0.68–0.88	7 (15%)	7 (16%)	4 (11%)	12 (34%)
Q3: 0.92–1.44	14 (30%)	11 (26%)	7 (19%)	8 (23%)
Q4: 1.47–7.11	7 (15%)	10 (23%)	4 (11%)	10(29%)

*CRTC3* rs12915189-nonGG: Q1 vs. 4 (reference), OR: 10.3 (95% CI 2.42–52.8); *p* = 0.003.

## Data Availability

The data presented in this study are available on request from the corresponding author.

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
