# Peer review of "Low Blood-As Levels and Selected Genotypes Appears to Be Promising Biomarkers for Occurrence of Colorectal Cancer in Women"

_biomedicines, 2021, doi:10.3390/biomedicines9091105_

Round 1

Reviewer 1 Report

Baszuk P and Stadnik P et al in this work analyzed the blood arsenic (As) levels combined with specific polymorphisms in MT1B, GSTP1, ABCB1, NQO1, CRTC3, GPX1, SOD2, CAT, XRCC1, ERCC2 for their association with the detection of colorectal cancer (CRC) among Polish women and found that low blood-As levels are significantly associated with increased CRC occurrence and that particular gene variants significantly enhanced this correlation. THE authors studied the female cases and controls, considering the male CRC incidence and mortality are well exceeded the females, what is the rationale to study the female CRC biomarkers? The authors need to discuss the results and how they may indicate to both sex populations. In addition, the molecular analysis section of genomic DNA genotyping did not reveal how the polymorphisms were identified.

  1. The authors need to keep pace with the recent publications, in lines 46-48, they cited GLOBOCAN 2018 data and the results, which are outdated. a new GLOBOCAN 2020 is available and the data are updated.
  2. The authors listed the smoking status in the female groups, did they the CRC occurrence is related to smoking?  Is smoking a possible way for arsenic exposure?

Author Response

Manuscript ID: biomedicines-1343455

Low blood-As levels and selected genotypes appears to be promising biomarkers for occurrence of colorectal cancer in women.

Editor and Reviewer comments:
Reviewer #1:

Baszuk P and Stadnik P et al in this work analyzed the blood arsenic (As) levels combined with specific polymorphisms in MT1B, GSTP1, ABCB1, NQO1, CRTC3, GPX1, SOD2, CAT, XRCC1, ERCC2 for their association with the detection of colorectal cancer (CRC) among Polish women and found that low blood-As levels are significantly associated with increased CRC occurrence and that particular gene variants significantly enhanced this correlation.

  1. The authors studied the female cases and controls, considering the male CRC incidence and mortality are well exceeded the females, what is the rationale to study the female CRC biomarkers?

Response:
Recently, we have been able to show that As is a major cancer risk factor for women in Poland – high levels of blood As (the highest quartile) is associated with more than 10-times higher risk of cancers as compared to quartile with the lowest As levels (Marciniak et al., Int J Cancer 2020). The above correlations have not been observed in ou studies of men (data not shown).

  1. The authors need to discuss the results and how they may indicate to both sex populations.

Response:
Recently, we recognized unexpectedly that one of mechanism of As induced carcinogenesis is via estrogens. Experimental studies on animals have shown dependence of As effects on sex (Huang  et al. Arch Toxicol. 2017).  In prospective cohorts we also observed that As associated cancer risk was correlated with hormonal status: - in women, higher blood As level was associated with higher cancer risk but - in men, especially at age <60 yrs, higher blood As level was associated with lower cancer risk. In females, As may influence on development of cancers by disrupting the function of estrogen receptors and suppressing the signaling pathway of estrogen (Gamble et al. Am J Clin Nutr 2007, Chatterjee et al. Reprod Biol Endocrinol. 2010). As is also a potential metallo-estrogen (Aquino et al. J Environ Sci Health C Environ Carcinog Ecotoxicol Rev., Ruiz-Ramos et al., Toxical Appl Pharmacol. 2009).

Actually, we are studing men and women. Our plan is to publish data on men separately.

  1. In addition, the molecular analysis section of genomic DNA genotyping did not reveal how the polymorphisms were identified.

Response:
Description of chosen polymorphisms is more detailed now.

4. The authors need to keep pace with the recent publications, in lines 46-48, they cited GLOBOCAN 2018 data and the results, which are outdated. a new GLOBOCAN 2020 is available and the data are updated.

Response:

Updated with the latest data: “According to GLOBOCAN 2020 data, colorectal cancer (CRC) was the third most frequently diagnosed cancer and was classified as one of the most common types of cancers in both men and women.”

5. The authors listed the smoking status in the female groups, did they the CRC occurrence is related to smoking?  Is smoking a possible way for arsenic exposure?

Response:
Participant have been matched for smoking.

Reviewer 2 Report

In the manuscript, the authors describe their findings in a case-control study of a retrospective cohort of women with colorectal adenocarcinoma. Specifically, the authors analyzed the blood arsenic levels of the cohort of women, some germline polymorphisms of a panel of genes involved in various processes, and finally calculated the OR of developing cancer in relation to arsenic levels and polymorphisms. The manuscript is well organized and clear, and the data convincing and potentially impactful for the prevention of this neoplasm.

Comments:
Why was a female-only cohort chosen? The authors need to justify this
The conclusions reported at line 164 of the manuscript on cancer cell metabolism and low blood arsenic levels in patients with adenocarcinoma is inappropriate. Being germinal, not somatic, polymorphisms belonging to the tumor, the metabolism is not of cancer cells, but in general of women.  
Is the distribution of disease stage across quartiles of As levels random, or are there differences?

Author Response

Manuscript ID: biomedicines-1343455

Low blood-As levels and selected genotypes appears to be promising biomarkers for occurrence of colorectal cancer in women.

Editor and Reviewer comments:
Reviewer #2:

In the manuscript, the authors describe their findings in a case-control study of a retrospective cohort of women with colorectal adenocarcinoma. Specifically, the authors analyzed the blood arsenic levels of the cohort of women, some germline polymorphisms of a panel of genes involved in various processes, and finally calculated the OR of developing cancer in relation to arsenic levels and polymorphisms. The manuscript is well organized and clear, and the data convincing and potentially impactful for the prevention of this neoplasm.

Comments:
1. Why was a female-only cohort chosen? The authors need to justify this.

Response:
Justification added in the introduction and discussion sections.

2.The conclusions reported at line 164 of the manuscript on cancer cell metabolism and low blood arsenic levels in patients with adenocarcinoma is inappropriate. Being germinal, not somatic, polymorphisms belonging to the tumor, the metabolism is not of cancer cells, but in general of women.  

Response:
Observed correlation may be also site-specific since in analogous studies performed in Our Centre, breast and lung cancer female patients did not show association between As level and tumour occurrence (data not shown). Mechanisms behind our observations need further investigations incuding somatic and genetic changes of various pathways involved in cancer development.
Recently, we recognized unexpectedly that one of mechanism of As induced carcinogenesis is via estrogens. Experimental studies on animals have shown dependence of As effects on sex (Huang  et al. Arch Toxicol. 2017).  In prospective cohorts we also observed that As associated cancer risk was correlated with hormonal status: - in women, higher blood As level was associated with higher cancer risk but - in men, especially at age <60 yrs, higher blood As level was associated with lower cancer risk. In females, As may influence on development of cancers by disrupting the function of estrogen receptors and suppressing the signaling pathway of estrogen (Gamble et al. Am J Clin Nutr 2007, Chatterjee et al. Reprod Biol Endocrinol. 2010). As is also a potential metallo-estrogen (Aquino et al. J Environ Sci Health C Environ Carcinog Ecotoxicol Rev., Ruiz-Ramos et al., Toxical Appl Pharmacol. 2009).

3.Is the distribution of disease stage across quartiles of As levels random, or are there differences?

Response:

Enclosed please find correlation between blood As levels and staging (Table).

n=83

Cases in stages

As blood level (µg/L)

stage I

stage II

stage III

stage IV

Q1: 0.27-0.67

6

19

13

1

Q2: 0.68-0.88

41

7

3

1

Q3: 0.92-1.44

4

8

8

1

Q4: 1.47-7.11

4

1

5

1

In our opinion differences/and numbers are not large enough to make appopriate conclusions.

Round 2

Reviewer 1 Report

All concerns are addressed.

Author Response

Dear Reviever,

Thank you for your review and taking the time to check our responses.

Yours sincerely,

Jan Lubiński

and co-authors

Szczecin, 24.08.2021